# Diagnostics of Secondary Fracture Properties Using Pressure Decline Data during the Post-Fracturing Soaking Process for Shale Gas Wells

**Jianfa Wu [1], Liming Ren [2], Cheng Chang [1], Shuyao Sheng [1], Jian Zhu [3], Sha Liu [1], Weiyang Xie [1] and Fei Wang [3,\***

[1] Shale Gas Research Institute, PetroChina Southwest Oil & Gas Field Company, Chengdu 610051, China; wu_jianfa@petrochina.com.cn (J.W.); chang_cheng@petrochina.com.cn (C.C.); shengsy_2022@petrochina.com.cn (S.S.); liusha63@petrochina.com.cn (S.L.); xieweiyang@petrochina.com.cn (W.X.)
[2] Advisory Center, PetroChina Southwest Oil & Gas Field Company, Chengdu 610051, China; renlm@petrochina.com.cn
[3] State Key Laboratory of Petroleum Resources and Prospecting, China University of Petroleum—Beijing, Beijing 102249, China; 2021310156@student.cup.edu.cn
**\*** Correspondence: wangfei@cup.edu.cn

**Abstract:** In addition to main fractures, a large number of secondary fractures are formed after the volumetric fracturing of shale gas wells. The secondary fracture properties are so complex, that it is difficult to identify and diagnose by direct monitoring methods. In this study, a new approach to model and diagnose secondary fracture properties is presented. First, a new pressure decline model, which is composed of four interconnected domains, i.e., wellbore, main fractures, secondary fractures, and reservoir matrix pores, is built. Then, the fracturing fluid pumping and post-fracturing soaking processes are simulated. The simulated pressure derivatives reflect five fracture-dominated flow regimes, which correspond to multiple alternating positive and negative slopes of the pressure decline derivative. The results of sensitivity simulation show that the density, permeability, and width of secondary fractures are the main controlling factors affecting the size ratio. Finally, based on the simulated pressure decline characteristics, a diagnostic method for the identification and analysis of secondary fracture properties is formed. This method is then applied to three platform wells in the Changning shale gas field in China. This study builds the correlation between the secondary fracture properties and the shut-in pressure decline characteristics, and also provides a theoretical method for comprehensive post-fracturing evaluation of shale gas horizontal wells.

**Keywords:** secondary fracture; soaking; shale gas; fracture diagnostics

## 1. Introduction

Shale gas reservoir development mostly uses multi-stage hydraulic fracturing technology in horizontal wells. Because natural fractures are a widespread growth in shale gas reservoirs, a large number of secondary fractures that communicate with the main fractures are formed after hydraulic fracturing. It is estimated that more than 60% of fracturing fluid is propped by secondary fractures in Woodford shale, in the Anadarko basin [1]. The properties of secondary fractures are more complex than that of natural fractures [2], because they have both storability and conductivity, leading to massive fracturing fluid storage and pressure diffusion. However, field monitoring tools for fracture geometries, like microseismic mapping, tracer tracking, and image logging, make it difficult to diagnose subtle differences in the properties of secondary fractures [3–5].

Based on well test theory, many pressure decline models for injection falloff tests have been proposed for post-fracturing flow regime identification. The representative work is the semi-log derivative plot of pressure decline with pump shutdown time proposed by Soliman in 1986 [6]. In 1989, Bourdet [7] calculated the slope of pressure derivatives of each

flow regime by using Horner time, Agarwal effective time, and superposition time. This method is only suitable for conventional reservoirs. In 2005, Soliman [8] drew the pressure decline derivative of fall-off test data with superposition time as the time axis and divided bilinear flow, linear flow, and radial flow according to the slope of the derivative curve.

Subsequently, a large number of studies emerged on the identification of fluid leak-off and fracture closure. Mohamed et al. [9] and Marongiu-Porcu et al. [10] proposed using Bourdert log–log special plots to identify normal leak-off and determine fracture closure. Bachman [11] et al. proposed a systematic standardized pressure decline analysis method based on log–log plots to identify various flow regimes before and after fracture closure. Liu [12] and Ehlig-Economides [13] proposed G-function-based analytical models to represent before-closure non-ideal leak-off behaviors. Wang [14] and Sharma proposed pressure decline analysis to characterize the properties of natural fractures and fracture stiffness for propped and un-propped fractures. In addition to the analytical models, there are some remarkable simulation studies. McClure et al. [15]. conducted rigorous simulation studies of various fracture compliance effects on before-closure pressure responses. Zanganeh et al. [16,17]. demonstrated progressive fracture closure behavior on various pressure decline special plots. Several mechanisms, like wellbore storage, tip extension, and residual fracture conductivity on the fall-off test data are evaluated. The above-summarized pressure decline models, including analytical and numerical models, pay more attention to the main fracture and natural fractures for the injection fall-off test data. Very few studies have attempted to investigate secondary fracture properties in volumetric fractured shale gas wells.

Between the fracturing fluid pumping treatment and well-opening flowback, the shale gas wells are usually shut in from 3 to 14 days for soaking. And that is a routine in the field for bottom-hole pressure diffusion and artificial fracture closure, preventing proppant backflow. During this soaking period, the wellhead-monitoring pressure continues to decline. However, this pressure decline data has not been fully utilized or underappreciated due to a lack of suitable models and methods, and that leads to difficulty in obtaining the unique properties of hydraulically fractured shale, i.e., secondary fracture properties. This study innovatively designs a pressure drop interpretation model during the well shut-in time and aims to build a correlation between the secondary fracture properties and the shut-in pressure decline characteristics. First, a post-fracturing shut-in pressure decline model is established on the basis of our previous proposed flowback model [18] by resetting the subprime grid and redefining the secondary fracture properties for numerical simulation. Then, simulation cases with different parameter combinations of secondary fracture properties are run and the corresponding characteristic curves of bottom-hole pressure decline are obtained. Based on the simulation results, a diagnosis method for secondary fracture properties is proposed. Finally, field case application is conducted by real data diagnosis and history matching analysis. It proves the proposed method can be helpful in an intensive explanation of artificial fractures in hydraulically fractured shale gas wells.

## 2. Pressure Decline Model during the Post-Fracturing Soaking Process

### 2.1. The Physical Model

Fracturing fluid is pumped step by step during the treatment of hydraulic fracturing in a shale gas well. After pumping at each stage, the pumping is stopped for several minutes to ten minutes. Once the final stage of pumping is completed, the pumping is stopped and the well is shut in for several days of soaking. The bottom-hole pressure decreases during the soaking periods, and the high pressure contained in the main and secondary fractures will be released into the reservoir. The fracturing fluid in the fracture system will leak into the matrix, and the natural gas in the matrix will be displaced into the fractures. Figure 1 displays the mass transfer diagram.

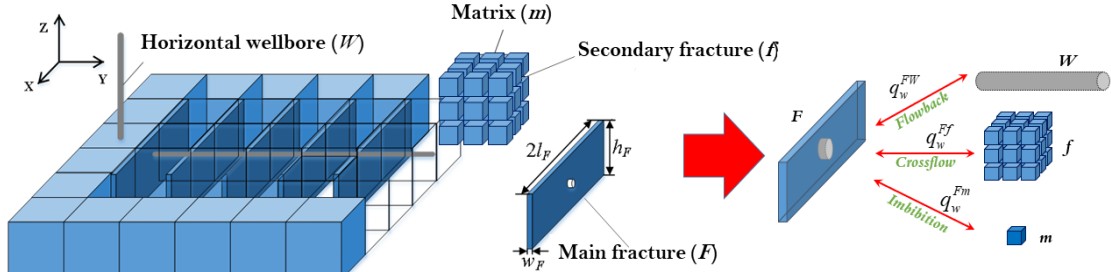

**Figure 1.** Schematic diagram of pressure decline model and mass transfer "matrix-secondary fracture-main fracture" during post-fracturing soaking process.

The physical model of a volumetric fractured shale gas well is simplified into four interconnected domains, i.e., horizontal wellbore (W), main fractures (F), secondary fractures (f), and matrix (m). The wellbore is connected to the main fracture, and the main fracture, secondary fracture, and matrix are connected in pairs. The three are continuously coupled through the flow and pressure at the contact surface. A schematic of the grid model design is shown in Figure 1, where X, Y, and Z represent the three directions of the grid model. The fracture networks, composed of main and secondary fractures, are composed of symmetrical, orthogonal vertical fractures of equal height. The grids of fractures are characterized by length, width, and permeability. The number of main fractures is equal to the number of perforation clusters. The density of secondary fractures is equal to the number of secondary fractures in the fracture stage area. The matrix is evenly distributed around the fracture network. Porosity and permeability are used to characterize the matrix storability and mobility, respectively. Wellbore is considered a source–sink term in the center of fracture networks.

*2.2. Mathematical Model*

2.2.1. Assumptions

Based on the above physical model, the assumptions for the mathematical model are as follows:

(1) The fracturing fluid is injected through the bottom hole, and the pumping process is considered to inject multiple stages at the same time. Each fracturing cluster creates one main fracture. The bridge plugs at all stages are completely dissolved and the wellbore is connected after the pumping treatment;

(2) The gas–water two-phase flow and isothermal flow are considered in this model. Fluid is slightly compressible;

(3) The gas flow in the main fractures is considered to be a high-velocity, non-Darcy flow. The gas flow in the secondary fractures is considered to obey Darcy flow conditions;

(4) Consider the compressibility of the main fractures, secondary fractures, and shale matrix;

(5) Consider matrix capillary imbibition.

2.2.2. Flow Conservation in Fractured Shale Reservoirs

Mass conservation equation of the water phase:

$$
\begin{aligned}
&\frac{\partial(\rho_w\phi^F S_w^F)}{\partial t} + \frac{\partial(\rho_w\phi^f S_w^f)}{\partial t} + \frac{\partial(\rho_w\phi^m S_w^m)}{\partial t} + \nabla\cdot\left[\rho_w\frac{k^F k_{rw}^F}{\eta_w}\nabla(p_w^F - \rho_w g D)\right] \\
&+ \nabla\cdot\left[\rho_w\frac{k^f k_{rw}^f}{\eta_w}\nabla(p_w^f - \rho_w g D)\right] + \nabla\cdot\left[\rho_w\frac{k^m k_{rw}^m}{\eta_w}\nabla(p_w^m - \rho_w g D)\right] = q_{sf}
\end{aligned}
\tag{1}
$$

where $\rho_w$ denotes the density of water (g/cm$^3$); $\phi^F$ denotes the porosity of the main fracture; $S_w^F$ denotes the water saturation of the main fracture; $\phi^f$ denotes the porosity of the secondary fracture; $S_w^f$ denotes the water saturation of the secondary fracture; $\phi^m$ denotes the porosity of the matrix; $S_w^m$ denotes the water saturation of the matrix; $k^F$ denotes

the absolute permeability of the main fracture; $k_{rw}^F$ denotes the relative permeability of the water phase of the main fracture; $p_w^F$ denotes the capillary pressure of the main fracture; $k^f$ denotes the absolute permeability of the secondary fracture; $k_{rw}^f$ denotes the relative permeability of the water phase of the secondary fracture; $p_w^f$ denotes the capillary pressure of the secondary fracture; $k^m$ denotes the absolute permeability of the matrix; $k_{rw}^m$ denotes the relative permeability of the water phase of the matrix; $p_w^m$ denotes the capillary pressure of the matrix; $\eta_w$ denotes the water mobility ratio; and $q_{sf}$ denotes the water injection rate $(g/cm^3 \cdot s)$.

Mass conservation equation of the gas phase:

$$
\begin{aligned}
&\frac{\partial(\rho_g \phi^F S_g^F)}{\partial t} + \frac{\partial(\rho_g \phi^f S_g^f)}{\partial t} + \frac{\partial(\rho_g \phi^m S_g^m)}{\partial t} + \nabla \cdot \left[ \lambda \rho_g \frac{k^F k_{rg}^F}{\eta_g} \nabla(p_g^F - \rho_g g D) \right] \\
&+ \nabla \cdot \left[ \rho_g \frac{k^f k_{rg}^f}{\eta_g} \nabla(p_g^f - \rho_g g D) \right] + \nabla \cdot \left[ \rho_g \frac{k^m k_{rg}^m}{\eta_g} \nabla(p_g^m - \rho_g g D) \right] = -\hat{q}_g
\end{aligned}
\tag{2}
$$

where $\rho_g$ denotes the density of gas $(g/cm^3)$; $S_g^F$ denotes the gas saturation of the main fracture; $S_g^f$ denotes the gas saturation of the secondary fracture; $S_g^m$ denotes the gas saturation of the matrix; $k_{rg}^F$ denotes the relative permeability of the gas phase of the main fracture; $p_g^F$ denotes the capillary pressure of the main fracture; $k_{rg}^f$ denotes the relative permeability of the gas phase of the secondary fracture; $p_g^f$ denotes the capillary pressure of the secondary fracture; $k_{rg}^m$ denotes the relative permeability of the gas phase of the matrix; $p_g^m$ denotes the capillary pressure of the matrix; $\eta_g$ denotes the gas mobility ratio; and $\hat{q}_g$ denotes the gas imbibition rate $(g/cm^3 \cdot s)$.

The correction factor $\lambda$ in Equation (2), which is used to correct the high-velocity, non-Darcy flow of gas in the main fracture, can be defined as [19]:

$$
\lambda = \frac{2}{1 + \sqrt{1 + 4\rho_g \beta \left( \frac{k^F k_{rg}^F}{\eta_g} \right)^2 |\nabla p^F|}}
\tag{3}
$$

where $\beta$ is the empirical coefficient of Forchheimer, which can be calculated by the following equation [19]:

$$
\beta = 3.2808 \times \frac{1.485 \times 10^9}{\left( k^F k_{rg}^F \times 10^{-15} \right)^{1.021}}
\tag{4}
$$

There exists fluid inflow and outflow between the two adjacent media of fractured reservoirs, but they cancel each other in the above conservation equation. The water phase flow rate between the triplet media can be expressed as follows:

$$
q_w^{Ff} = \frac{\alpha_2 \rho_w k^f k_{rw}^f (p_w^F - p_w^f)}{\eta_w}
\tag{5}
$$

where $q_w^{Ff}$ denotes the water flow rate from the main fracture to the secondary fracture $(g/cm^3)$;

$$
q_w^{Fm} = \frac{\alpha_3 \rho_w k^m k_{rw}^m (p_w^F - p_w^m)}{\eta_w}
\tag{6}
$$

where $q_w^{Fm}$ denotes the water flow rate from the main fracture to the matrix $(g/cm^3)$; and

$$
q_w^{fm} = \frac{\alpha_4 \rho_w k^m k_{rw}^m (p_w^f - p_w^m)}{\eta_w}
\tag{7}
$$

where $q_w^{fm}$ denotes the water flow rate from the secondary fracture to the matrix $(g/cm^3)$.

Considering the compressibility of fractures and matrix pores, a supplementary pressure-dependent equation for porosity and permeability is required:

$$\phi^{F/f/m} = \phi_0^{F/f/m} e^{C_\phi^{F/f/m}(p_g^{F/f/m} - p_{g0})} \tag{8}$$

$$k^{F/f/m} = k_0^{F/f/m} e^{d^{F/f/m}(p_g^{F/f/m} - p_{g0})} \tag{9}$$

Considering the compressibility of gas and water, a supplementary pressure-dependent equation for density is required:

$$\rho_w = \rho_{w0} e^{C_w(p_w^{F/f/m} - p_{w0}^{F/f/m})} \tag{10}$$

$$\rho_g = \rho_{g0} e^{C_g(p_g^{F/f/m} - p_{g0}^{F/f/m})} \tag{11}$$

where $\rho_w$ denotes the density of water (g/cm$^3$) and $\rho_g$ denotes the density of gas (g/cm$^3$).

The relationship between fracture width and porosity is given using the Carman–Kozeny equation [20]:

$$\phi^f = nwb \tag{12}$$

Considering the two-phase flow of gas and water in the fracture and matrix, a supplementary constraint equation for water saturation is required:

$$S_w^{F/f/m} + S_g^{F/f/m} = 1 \tag{13}$$

The fluid compressibility equation is:

$$\begin{cases} \rho_w = \rho_w e^{C_w(p_w^{F,f,m} - p_{w0}^{F,f,m})} \\ \rho_g^{-1} = \rho_{g0}^{-1} \left[ 1 + C_g \left( p_g^{F,f,m} - p_{g0}^{F,f,m} \right) \right] \end{cases} \tag{14}$$

In addition, due to the relatively high permeability of the fracture network, the capillary force is approximately zero, and the matrix has capillary force, which can be expressed as:

$$p_g^{F/f} = p_w^{F/f} \tag{15}$$

$$p_g^m - p_w^m = p_c^m \tag{16}$$

### 2.2.3. Initial and Boundary Conditions

The selected simulation unit satisfies the closed outer boundary conditions. The bottom-hole flow pressure is calculated according to the fluid pumping equation of hydraulic fracturing as follows:

$$\hat{q}_{sf} = \frac{\alpha_1 \rho_w k^F k_{rw} \left( p_{wf} - p_w^F \right)}{\eta_w B_w} = \frac{2\pi k^F k_{rw}^F \rho_w}{\eta_w ln\left( \frac{r_e}{r_w} + S \right) B_w \Delta x \Delta z} \left( p_{wf} - p_w^F \right) \tag{17}$$

where $B_w$ denotes the water phase volume factor (cm$^3$/cm$^3$); $q_{sf}$ denotes the water injection rate (g/cm$^3$·s); $p_{wf}$ denotes the bottom-hole pressure ($10^{-1}$ MPa); $k^F$ denotes the absolute permeability of the main fractures (µm$^2$); $k_{rw}^F$ denotes the water phase relative permeability of the main fractures; $\rho_w$ denotes the density of water (g/cm$^3$); $\eta_w$ denotes the viscosity of water (mPa·s); $r_e$ denotes the supply radius (cm); $r_w$ denotes the well radius (cm); $S$ denotes the skin factor; and $\Delta x$ and $\Delta z$ denote the grid size of different directions for the numerical model.

The initial pressure and water saturation of the matrix and fracture are assumed to be the same in the original, undeveloped reservoir state. The pumping stage of fracturing is simulated as a constant bottom-hole pressure injection. By adjusting the

pressure-dependent porosity and permeability coefficients of the fracture and matrix (Equations (8) and (9)), the injection rate and injection volume of fracturing fluid in the model is consistent with the actual fracturing treatment. At the end of pumping, the pressure field and water saturation distribution obtained are used as the initial conditions to simulate the shut-in pressure decline, which can be expressed as follows:

$$p_w^F(x,y,z,t)\Big|_{t=0} = p_{wi}^F \tag{18}$$

$$S_w^F(x,y,z,t)\Big|_{t=0} = S_{wi}^F \tag{19}$$

$$p_w^f(x,y,z,t)\Big|_{t=0} = p_{wi}^f \tag{20}$$

$$S_w^f(x,y,z,t)\Big|_{t=0} = S_{wi}^f \tag{21}$$

$$p_w^m(x,y,z,t)\big|_{t=0} = p_{wi}^m \tag{22}$$

$$S_w^m(x,y,z,t)\big|_{t=0} = S_{wi}^m \tag{23}$$

In this model, the finite difference method is used in the discretization of the partial differential equations of the simulation of a shale gas well after fracturing, the semi-implicit method is used to deal with the nonlinear equations, and the Newton–Raphson iterative method is used to solve the equations.

## 3. Numerical Simulation Method

The numerical model was established according to the geological and construction parameters of a fractured horizontal well in a typical shale gas reservoir. The initial reservoir pressure was set to 45 MPa, the length, width, and thickness were 2000, 560, and 30 m, respectively, and a hydraulically fractured horizontal well with a lateral length of 1800 m was located in the center of a shale reservoir. The total fracture stages were 30. In every stage, there were six identical, transverse, hydraulic fractures with a fracture half-length of 135 m, according to the normal fracturing operations and Microseismic monitoring results in a shale gas field in South China. The input reservoir and hydraulic fracture parameters of the simulation model are listed in Table 1.

In this model, gas–water relative permeability and capillary pressure in shale matrix were set according to the core experimental data [21,22]. The parameters related to the seepage property of the main and secondary fractures were set refer to the typical flowback model [23,24] of hydraulically fractured shale gas reservoirs. The fracturing pumping was simulated as a water injection process, and the total pumping volume was 45,300 m$^3$. After the 10 d injection, the well was shut in for 80 days for soaking up.

**Table 1.** The input reservoir and hydraulic fracture parameters of the simulation model.

| Variable, Symbol | Value | Variable, Symbol | Value |
|---|---|---|---|
| Main fracture half-length | 135 m | Rock compressibility | $4.4 \times 10^{-4}$ MPa$^{-1}$ |
| Main fracture conductivity | 8 D·cm | Gas compressibility | 0.03 MPa$^{-1}$ |
| Main fracture porosity | 0.3 | Fracturing fluid viscosity | 0.8 mPa·s |
| Matrix permeability | $7 \times 10^{-4}$ mD | Fracturing fluid density | 1000 kg/m$^3$ |
| Matrix porosity | 0.06 | Initial water saturation | 0.35 |
| Secondary fracture permeability [23] | 0.01 mD | Initial reservoir pressure | 45 MPa |
| Secondary fracture porosity [24] | 0.055 | Gas viscosity | 0.058 mPa·s |
| Fracturing fluid compressibility | $4.8 \times 10^{-7}$ MPa$^{-1}$ | Secondary fracture closure coefficient [24] | 0.014 MPa$^{-1}$ |
| Main fracture closure coefficient [24] | 0.0087 MPa$^{-1}$ | Secondary fracture density [24] | 3 m$^{-2}$ |

## 4. Results and Discussion

### 4.1. Simulation Results of Bottom-Hole Pressure Decline Characteristics

Figure 2 displays the simulated bottom-hole flowing data. The pressure decline and derivative of pressure decline defined by Bourdet et al. [7,8] were used to draw a double logarithmic characteristic curve, as shown in Figure 2, that presents the pressure decline derivative. There are five characteristic slope segments with positive and negative links. It can be seen that stage ① is controlled by the main fracture, which is in the earliest stage and has the fastest pressure decline rate; the first V-shape (stages ② and ③) is controlled by the secondary fracture, which is in the middle stage of the soaking and the pressure decline rate slows down. The second V-shape (stages ④ and ⑤) is controlled by the secondary fracture and matrix, which is in the late stage of the soaking and has a slow pressure decline rate.

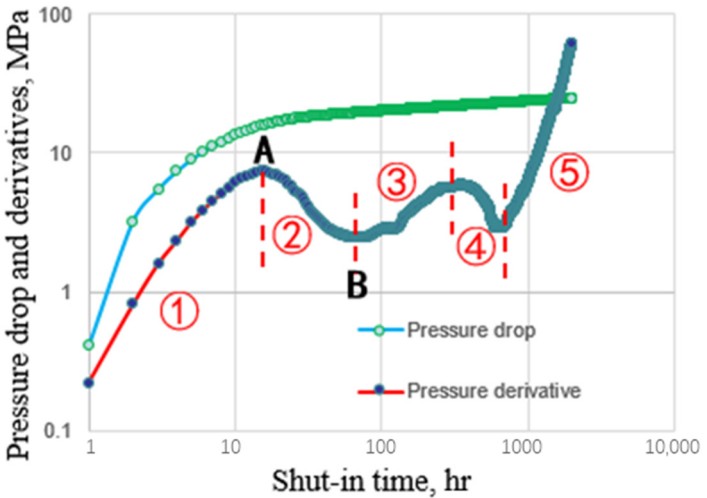

**Figure 2.** Simulated pressure decline and derivatives on log–log plot.

### 4.2. Comparison of Pressure Decline Characteristic Curves

In order to identify the characteristics of pressure decline during the soaking process, the model was used to carry out five sets of single-factor sensitivity simulations and combined-factor simulations for the main and secondary fractures.

Figure 3a shows the length of the main fracture, which determines the duration of stage ① and the pressure decline derivative value at point A. The larger the length of the main fracture is, the larger the time and derivative value at point A is, and the higher the overall level of pressure decline derivative value is in the later stage. Figure 3b shows that the conductivity of the main fracture has a weak influence on the shape of the pressure decline derivative.

Figure 4 illustrates that properties of secondary fractures determine the size ratio and concave–convex degree of the two V-shapes of the sawtooth-shaped pressure decline characteristic curve. Figure 4a shows that the density of secondary fractures determines the depth of the V-shapes. The higher the density of the secondary fracture is, the deeper the V is. Figure 4b shows that the width of secondary fractures determines the concave–convex degree of the two V-shapes. The wider the width of the secondary fracture, the shallower the V-shape is. The V-shape reduces one log cycle. Figure 4c shows the permeability of the secondary fracture determines the duration of stage ③ and the pressure decline derivative value at point B. The larger the conductivity of the secondary fracture, the longer the duration of stage ③, and the duration stage is prolonged 1–1.5 log cycles. The smaller corresponding time of the derivative value is at point B.

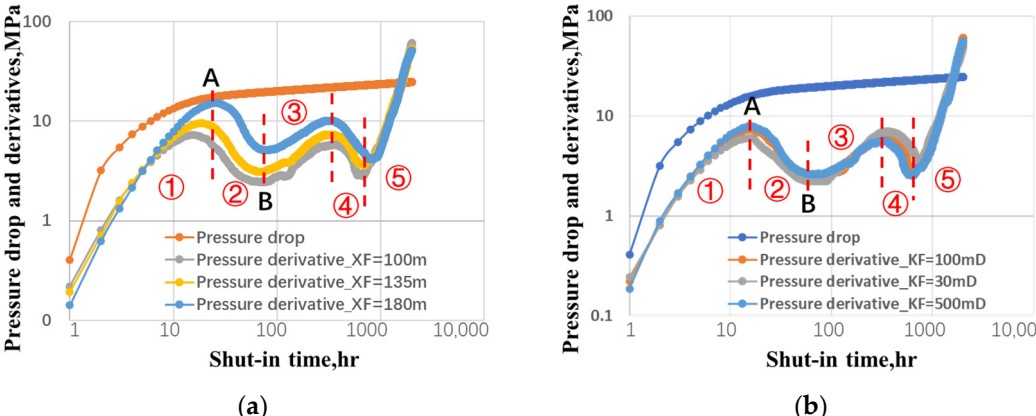

**Figure 3.** Characteristic curves comparison of main fracture properties: (**a**) main-fracture half-length; (**b**) main-fracture conductivity.

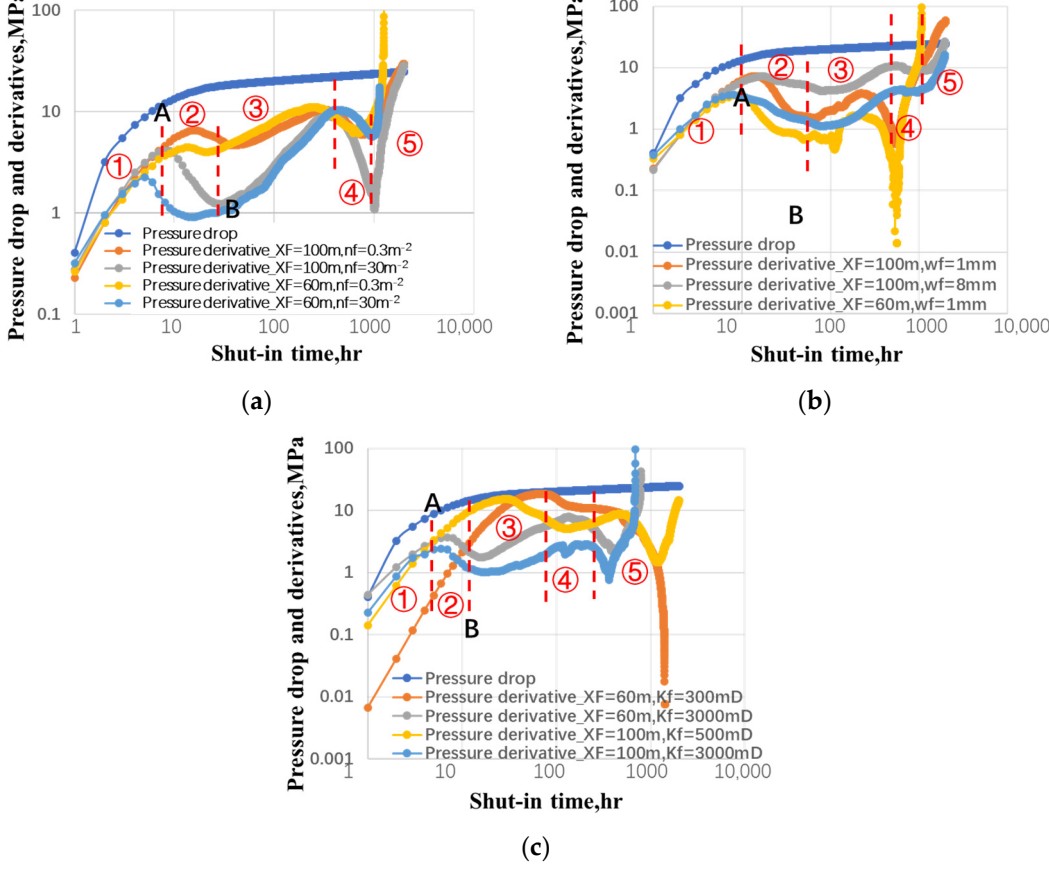

**Figure 4.** Characteristic curves comparison of secondary fracture properties: (**a**) secondary-fracture density; (**b**) secondary-fracture width; (**c**) secondary-fracture permeability.

### 4.3. Diagnostic Method Based on Simulated Pressure Decline Derivatives

A diagnostic method based on simulated pressure decline derivatives can be established for the hydraulically fractured horizontal wells in the same reservoir. There are three steps, which are as follows:

Step 1: The dimensionless pressure decline is plotted with the unit fracturing treatment scale, including the fracturing fluid volume and stage numbers, according to the actual monitoring bottom-hole or wellhead pressure data. The absolute value of the dimensionless

pressure decline determines the fracture size. The larger the absolute pressure decline value, the larger the fracture size;

Step 2: The pressure decline and its derivative are plotted on a log–log plot. Based on the characteristic curves in Figures 3 and 4, the main fracture length can be determined according to the pressure decline derivative value at point A and the overall level of pressure decline derivative value. The density, width, and permeability of secondary fractures can be diagnosed by the size ratio and concave–convex degree of the two V-shapes of the pressure decline characteristic curve;

Step 3: Based on the above qualitative diagnosis, the quantitative calculation of fracture properties requires the matching of the pressure decline history during soaking periods.

## 5. Field Case Study

### 5.1. Geological and Construction Overview of the H49 Platform

The H49 Platform is located in the Changning shale gas reservoir in Southwest China. The reservoir is the Longmaxi Formation shale, with relatively developed natural fractures and high brittleness, but a large stress difference. The initial reservoir pressure was 45 MPa, matrix permeability was 0.0005–0.0012 mD, porosity was 0.07, and gas saturation was 0.65. The H49 Platform deployed three horizontal wells—H49-6, H49-7, and H49-8—with 300 m spacing. The fracturing construction amount of each well is shown in Table 2.

**Table 2.** Fracturing construction parameters of H49 Platform.

| Well | Vertical Depth (m) | Lateral Length (m) | Number of Stages | Number of Clusters | Fluid Volume (m³) | Proppant Volume (t) | Fracturing Time (d) | Shut-in Time (h) |
|------|------|------|------|------|------|------|------|------|
| H49-6 | 2798 | 1840 | 28 | 192 | 48,280 | 4172 | 50 | 599 |
| H49-7 | 2814 | 1850 | 27 | 228 | 48,374 | 3820 | 27 | 611 |
| H49-8 | 2857 | 2064 | 32 | 254 | 57,808 | 5348 | 14 | 832 |

### 5.2. Diagnostics of Hydraulic Fracture Properties

Three wells on the H49 Platform were continuously monitored for wellhead pressure in each well during soaking. As shown in Figure 5, well H49-8 has the largest vertical depth and the longest duration of the initial pressure drop among the three wells, indicating the largest fracture scale. In wells H49-6 and H49-8, the wellhead pressure drop gradient trends are consistent at less than 0.04 MPa/h after one day of well soaking, indicating that the reservoir properties communicated by the fractures in both wells are consistent. Well H49-7 has a relatively large vertical depth and high-pressure location. Well H49-7 has the largest average pressure drop gradient, which is greater than 0.04 MPa/h, indicating that well H49-7 has a better conductivity of secondary fractures.

The pressure declines and derivatives of the three platform wells are shown in Figure 6, and it can be seen that the characteristic curves of both wells H49-6 and H49-8 show a sawtooth pattern. The characteristic stages ②–⑤ are more uniform and obvious, with more developed secondary fractures and a high degree of fracture network. Among them, stage ① of well H49-6 is short in duration and low in position, indicating the small scale of the main fractures, while well H49-8, in contrast to well H49-6, has a long duration and high overall position of stage ①, indicating a large number of main fractures, which is consistent with the clusters of fracturing construction. As a side well, well H49-8 is 600 m away from well H49-6. Its fractured main fracture is still in series with well H49-6, indicating that the main fracture extends well. This well still forms a complex fracture network. Stages ② and ③ of well H49-7 are not obvious, indicating that its number of secondary fractures is small. It can be seen that it may be related to the fact that it was fractured last in the middle of wells H49-6 and H49-8 and was affected by the stress field. Stages ④ and ⑤ of well H49-7 are very deeply V-shaped, indicating that the width and permeability of its generated secondary fractures are very small.

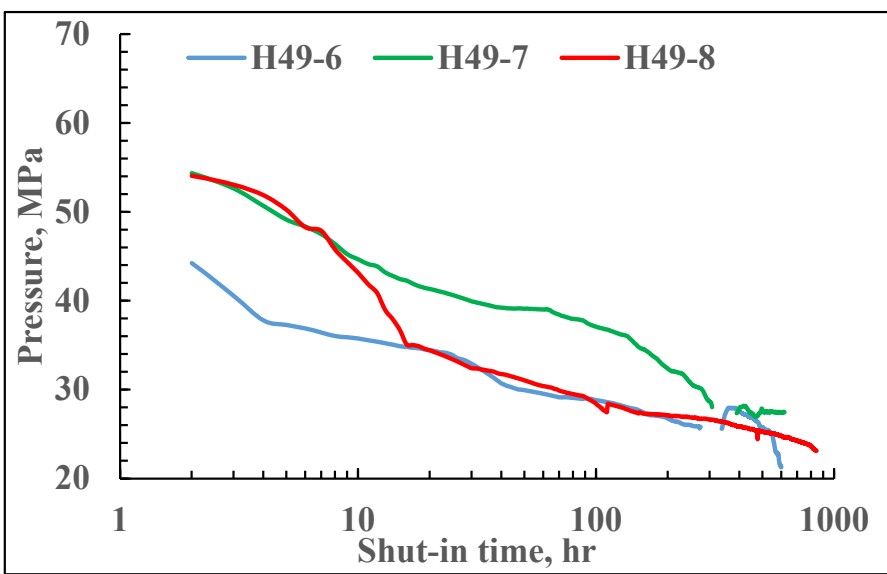

**Figure 5.** Wellhead monitoring pressure in H49 Platform during soaking process.

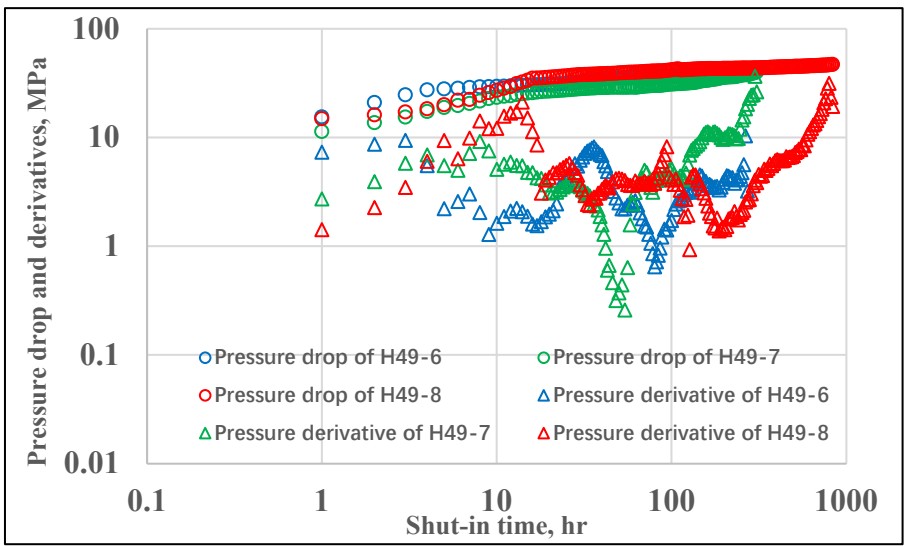

**Figure 6.** Pressure decline and derivatives of H49 Platform on log–log plot.

*5.3. Calculation of Hydraulic Fracture Properties*

The proposed pressure decline model was used to fit the soaking pressure drop history of three platform wells. Based on the above qualitative diagnosis, the quantitative calculation of fracture properties requires the matching of the pressure decline history during soaking periods, as shown in Figure 7. The fracture parameters obtained from the history matching are shown in Table 3. The main fracture half-length is 62–90 m, which is less than 150 m from the half-well distance, proving that the current fracturing treatment cannot meet the requirement of the reservoir plane utilization. The density and permeability of secondary fracture density are 3.16–4.93 $m^{-2}$ and 0.05–0.08 mD, with a given secondary fracture width of 2 mm.

As shown in Figure 8, the microseismic monitoring results gained from the field test show the average half-lengths of the hydraulic fractures of the three wells in the H49 Platform are 116.5 m, 125 m, and 130.7 m, respectively, and the average fracture network widths for each stage are 61 m, 48 m, and 75 m, respectively. Through comparison, it can be found that the ratio of the main fracture half-length and secondary fracture density explained by our proposed model to the microseismic monitoring results is consistent. Although due to differences in interpretation principles [25,26] and further fracture closure

during the soaking stage, the consistent interpretation results demonstrate that the fracture network parameters explained by the proposed method can reflect the actual fracturing effect.

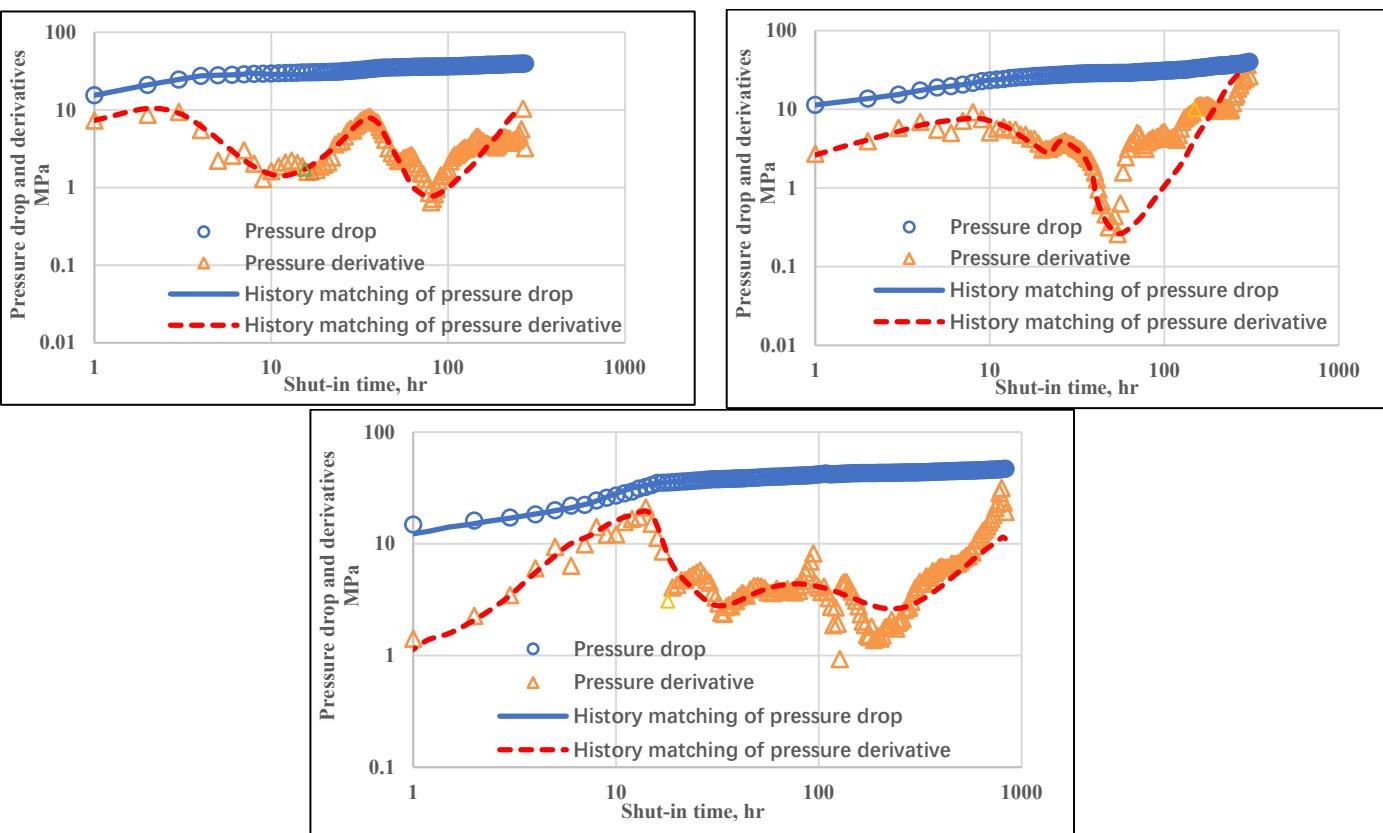

**Figure 7.** History matching of pressure decline in H49 Platform.

**Table 3.** Obtained hydraulic fracture properties.

| Parameters of Fracture Properties | H49-6 | H49-7 | H49-8 |
| --- | --- | --- | --- |
| Main fracture half-length (m) | $62 \pm 2.5$ | $70 \pm 2.5$ | $90 \pm 2.5$ |
| Main fracture conductivity (D·cm) | 9 | 10 | 9 |
| Secondary fracture permeability (mD) | 0.08 | 0.05 | 0.07 |
| Secondary fracture density ($m^{-2}$) | 4.21 | 3.16 | 4.93 |

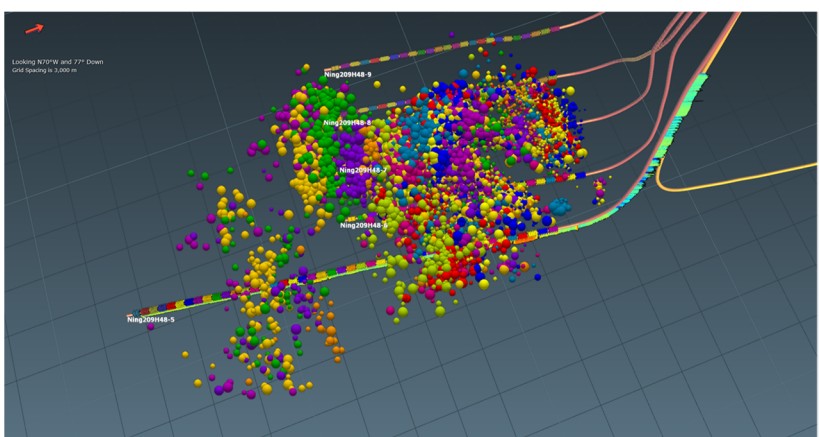

**Figure 8.** Microseismic monitoring results in H49 Platform. The red arrow represents the direction of vision, the color dots represent the microseismic events.

### 6. Conclusions

1.  A wellbore fracture–network gas reservoir coupled fracturing shut-in pressure decline model is proposed. The simulated pressure derivatives show a "sawtooth" shape on a log–log plot, reflecting five fracture-dominated flow stages. Among them, stage ① is controlled by the main fracture, which is in the earliest stage and has the fastest pressure decline rate; the first V-shape (stages ② and ③) is controlled by the secondary fracture, which is in the middle stage of the soaking and the pressure decline rate slows down; and the second V-shape (stages ④ and ⑤) is controlled by the secondary fracture and matrix, which is in the late stage of the soaking and has a slow pressure decline rate;

2.  The sensitivity simulation results show that the length of the main fracture determines the duration of stage ① and the pressure decline derivative value at point A. While the conductivity of the main fracture has a weak influence on the shape of the pressure decline derivative, the density, width, and permeability of the secondary fractures determine the size ratio and concave–convex degree of the two V-shapes of the pressure decline characteristic curve;

3.  Based on the pressure decline simulation, a diagnostic method is established for analyzing the pressure decline data and calculation of the main and secondary fracture properties of hydraulically fractured shale gas wells;

4.  A field case application proves that the proposed method works well for platform wells. For the H49 Platform, it indicates that the extension of both the main fracture and secondary fracture is better than that of central wells. The secondary fracture of central wells is limited, so it intends to generate simple main fractures.

**Author Contributions:** Methodology, J.W.; Software, L.R.; Validation, C.C.; Formal analysis, S.S. and F.W.; Data curation, J.Z. and S.L.; Writing—original draft, W.X. and F.W.; Funding acquisition, F.W. All authors have read and agreed to the published version of the manuscript.

**Funding:** This research was funded by the National Natural Science Foundation of China (No. 51974332).

**Data Availability Statement:** The data presented in this study are available on request from the corresponding author. The data are not publicly available due to data from oilfield confidential documents.

**Conflicts of Interest:** Authors Jianfa Wu, Liming Ren, Cheng Chang, Shuyao Sheng, Sha Liu, and Weiyang Xie were employed by China National Petroleum Corporation Southwest Oil and Gas Field. The remaining authors declare that the research was conducted in the absence of any commercial or financial relationships that could be construed as a potential conflict of interest.

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
