# Peer review of "Diagnostics of Secondary Fracture Properties Using Pressure Decline Data during the Post-Fracturing Soaking Process for Shale Gas Wells"

_processes, doi:10.3390/pr12020239_

Round 1

Reviewer 1 Report

Comments and Suggestions for Authors

The paper establishes a new model and analysis method for shale gas during the post-fracturing soaking process and gas-water two-phase flow. According to the shape displayed on the logarithmic graph by the simulated pressure derivatives, different dominant flow states of secondary fractures are reflected. The paper has a clear overall structure and a logical flow of content. This paper can be accepted after minor revision, please see the following comments.

1. The introduction does not specifically highlight the research innovation of the proposed method, please modify it.

2. Some Figures should be double-checked and refined to meet the required level of this journal. For example, in Figure 2, the red wavy line below ‘hr’.

3 In chapter 3.3, the author mentioned, " The larger the absolute pressure decline value, the larger the fracture size.” Please specify the corresponding relationship.

4 The detailed solution process can be removed since the semi-implicit finite difference solution is well known.

Author Response

1.The introduction does not specifically highlight the research innovation of the proposed method, please modify it.

The innovation points have been added in the introduction section.

  1. Some Figures should be double-checked and refined to meet the required level of this journal. For example, in Figure 2, the red wavy line below ‘hr’.

The red wavy line in the image has been modified.

3 In chapter 3.3, the author mentioned, " The larger the absolute pressure decline value, the larger the fracture size.” Please specify the corresponding relationship.

The corresponding relationship has been added in line 274.

4 The detailed solution process can be removed since the semi-implicit finite difference solution is well known.

The detailed solution process has been removed.

Reviewer 2 Report

Comments and Suggestions for Authors

The reviewed paper “Diagnostics of Secondary Fracture Properties using Pressure Decline Data during Post-fracturing Soaking Process for Shale Gas Wells” consider a possibility to estimate secondary hydraulic fracture properties with the help of pressure decline data after the injection stop. The suggested approach is applied to data obtained at three wells in Changning shale gas field, China.

The results are of practical interest, some comments are following.

1.     All used variables should be determined directly after the equations.

2.     Line 163 – what is “injection equation”?

3.     I did not find the gas state equation, do you use it?

4.     Shut-in time in your figures looks unreal, the fracturing durations are hours, shut-in time is minutes, it could not last for several months (10000 hours = 416 days, that is more than year)!

5.     Graphs are overlapped by legends, the figures should be changed.

6.     As a rule, the inverse problem (like considered in the paper) is not well posted, so it is necessary to discuss reliability of the obtained estimations.

I recommend thoroughly editing the text by native English speaking.

Comments on the Quality of English Language

A lot of typos in the text, some of them are marked by yellow.

Author Response

  1. All used variables should be determined directly after the equations.

The variables have been specified below the equations.

  1. Line 163 – what is “injection equation”?

The injection equation means the “fracturing-fluid pumping equation”, which has been modified the on line 189.

  1. I did not find the gas state equation, do you use it?

We added gas and water compressibility equations in equation (14).

  1. Shut-in time in your figures looks unreal, the fracturing durations are hours, shut-in time is minutes, it could not last for several months (10000 hours = 416 days, that is more than year)!

In this study, we focus on the whole well post-fracturing soaking process, which is the shut-in duration from the pumping stop time of the last fracturing stage to the whole well open to produce. Actually, the well post-fracturing soaking process usually takes several weeks in China. In this study, our simulation case is for shut in 80 days (1920 hours) for soaking up. We believe that is reasonable for a sufficient soaking process and pressure decline data analysis.

  1. Graphs are overlapped by legends, the figures should be changed.

The figures have been changed.

  1. As a rule, the inverse problem (like considered in the paper) is not well posted, so it is necessary to discuss reliability of the obtained estimations.

The two inversed parameters, i.e. the main-fracture half-length and secondary-fracture density have been compared with the micro-seismic monitoring results from field. Please find the text from line 346 to 356.

Reviewer 3 Report

Comments and Suggestions for Authors

It would be advisable to improve the given study, taking into account the specific recommendations:

1)[Line 11]: The abstract is too long – based on journal rules it should be a total of about 200 words maximum.

2) [Line 45/46]: The text transfer is not accurate.

3) [Line 97/98]: A space is required between the text and the notation in parentheses.

4) [Line 98]: Space between point and “The” is needed.

5) [Line 111]: Please precise title of the figure 1. You write – “Fig.1.displays the mass transfer diagram.” (Line 95); “Schematic of grid model as shown in Fig.1” (Line 100); and “Figure 1. Schematic diagram of pressure decline model and mass transfer during post-fracturing soaking process.” (Line 111).

6) Based on instruction for authors of journal Processes, section “Materials and methods” is missing.

7) [Line 155]: Please explain in which program the given model was created?

8) [Line 119/120/122/123]: Sentences are not usually started with parentheses. It would be necessary to make appropriate corrections in the structure of the sentences.

9) [Line 127]: Formula numbers are too large on the background of the overall text. Adjustments would be necessary.

10) Table 3: Please specify the unit of density.

11) [Line 221]: Space between point and “Upwind” is needed. Similarly, is also in Line 222 – no space between words.

12) [Line 231]: A parenthesis is not necessary at the beginning of a sentence.

13) [Line 233]: Space between point and “The” is needed.

14) [Line 257/260]: It is recommended to divide the sentence to make the information easier to understand.

15) [Line 286]: A more detailed analysis of the graphs shown in Figure 4 would be necessary.

16) [Line 320]: Shouldn't Table 2 be mentioned here?

17) [Line 339/344]: It is recommended to divide the sentence to make the information easier to understand.

18) [Line 347]: A more detailed analysis of the graphs shown in Figure 7 would be necessary.

19) [Line 363]: There is no space between table and the text.

20) [Line 363]: Please explain how these microseismic monitoring results were obtained?

21) State the scientific value (importance) of your contribution in the conclusions.

22) The presentation of the record of literature sources is different and all of them should be adjusted according to journal rules.

23) It is necessary to read the article thoroughly and make corrections in English. There are grammar mistakes in the text.

Comments on the Quality of English Language

Extensive editing of English language required. There are grammar mistakes in the text.

Author Response

1)[Line 11]: The abstract is too long – based on journal rules it should be a total of about 200 words maximum.

Summary has been modified in 200 words.

2) [Line 45/46]: The text transfer is not accurate.

The text has been revised in line 41.

3) [Line 97/98]: A space is required between the text and the notation in parentheses.

The corresponding parts of the text have been modified in line94/95.

4) [Line 98]: Space between point and “The” is needed.

The corresponding parts of the text have been modified in line95.

5) [Line 111]: Please precise title of the figure 1. You write – “Fig.1.displays the mass transfer diagram.” (Line 95); “Schematic of grid model as shown in Fig.1” (Line 100); and “Figure 1. Schematic diagram of pressure decline model and mass transfer during post-fracturing soaking process.” (Line 111).

The title of the figure 1 have been modified in line108

6) Based on instruction for authors of journal Processes, section “Materials and methods” is missing.

We modified the section 3 “Numerical Simulation Method”.

7) [Line 155]: Please explain in which program the given model was created?

The reference paper [19] has been marked in Line 154.

8) [Line 119/120/122/123]: Sentences are not usually started with parentheses. It would be necessary to make appropriate corrections in the structure of the sentences.

The corresponding parts of the text have been modified in line112-124.

9) [Line 127]: Formula numbers are too large on the background of the overall text. Adjustments would be necessary.

The formula number size has been reduced

10) Table 3: Please specify the unit of density.

The unit of density has been specified.

11) [Line 221]: Space between point and “Upwind” is needed. Similarly, is also in Line 222 – no space between words.

The text has been deleted.

12) [Line 231]: A parenthesis is not necessary at the beginning of a sentence.

The text has been deleted.

13) [Line 233]: Space between point and “The” is needed.

The text has been deleted.

14) [Line 257/260]: It is recommended to divide the sentence to make the information easier to understand.

The corresponding parts of the text have been modified in line240-244.

15) [Line 286]: A more detailed analysis of the graphs shown in Figure 4 would be necessary.

A more detailed analysis of Figure 4 has been added. Please find the text in from line 265 to 275.

16) [Line 320]: Shouldn't Table 2 be mentioned here?

The table 2 is modified.

17) [Line 339/344]: It is recommended to divide the sentence to make the information easier to understand.

The corresponding parts of the text have been modified in line317-324.

18) [Line 347]: A more detailed analysis of the graphs shown in Figure 7 would be necessary.

The more detailed analysis of the graphs has been added in line329-330

19) [Line 363]: There is no space between table and the text.

The space has been added between table and the text.

20) [Line 363]: Please explain how these microseismic monitoring results were obtained?

The microseismic monitoring results is from field test.

21) State the scientific value (importance) of your contribution in the conclusions.

The scientific value has been added in the conclusions.

22) The presentation of the record of literature sources is different and all of them should be adjusted according to journal rules.

The presentation of the record of literature sources has been adjusted.

23) It is necessary to read the article thoroughly and make corrections in English. There are grammar mistakes in the text.

Yes. We corrected all the mistakes we found.

Round 2

Reviewer 2 Report

Comments and Suggestions for Authors

Authors made efforts to improve the manuscript, but some improvements should still be made.

1. Graphs are still overlapped by legends, the figures 4, 6, 7 should be improved!

2. There are unresolved questions about the possibility of applying the approach based on the Bourdet method to the pressure drop curve obtained during multi-stage hydraulic fracturing.

3. On page 10 the numbering of sections is incorrect.

4. The duration of shut-in time in Fig. 5 does not correspond to the shut-in time in Table 2.

5. It is necessary to provide an assessment of the accuracy of the results obtained, for example: the length of the fracture is estimated to be 62+-1 meters, or something like that.

Comments on the Quality of English Language

Please check the English language thoroughly. For example, in abstract: In this study, a new approach to model and diagnose secondary fracture properties - there is no verb in this sentence.

Author Response

1. Graphs are still overlapped by legends, the figures 4, 6, 7 should be improved!

Figure 4,6,7 has been modified as required.

2. There are unresolved questions about the possibility of applying the approach based on the Bourdet method to the pressure drop curve obtained during multi-stage hydraulic fracturing.

 The possibility of applying the approach can be referenced by “Pump-stopping pressure drop model considering transient leak-off of fracture network. Petroleum Exploration and Development. 2023, 50 (02): 416-423.”

3. On page 10 the numbering of sections is incorrect.

The numbering of sections has been modified.

4. The duration of shut-in time in Fig. 5 does not correspond to the shut-in time in Table 2.

The duration of shut-in time in Table2 has been corrected.

5. It is necessary to provide an assessment of the accuracy of the results obtained, for example: the length of the fracture is estimated to be 62+-1 meters, or something like that.

The accuracy of the results obtained has been modified as required.

Reviewer 3 Report

Comments and Suggestions for Authors

Thanks for the improvements. Suggestions have been taken into account.

Comments on the Quality of English Language

Minor editing of English language required.

Author Response

Thank for reviewing the manuscript.